# Beyond the Limits of Shannon’s Information in Quantum Key Distribution

**DOI:** 10.3390/e23020229

**Published:** 2021-02-16

**Authors:** Luis Adrián Lizama-Pérez, J. Mauricio López R., Emmanuel H. Samperio

**Affiliations:** 1Dirección de Investigación, Innovación y Posgrado, Universidad Politécnica de Pachuca, Ex-Hacienda de Santa Bárbara, Zempoala, Hidalgo 43830, Mexico; esamperio593@micorreo.upp.edu.mx; 2Cinvestav Querétaro, Libramiento Norponiente 2000, Real de Juriquilla, Santiago de Querétaro, Querétaro 76230, Mexico; jm.lopez@cinvestav.mx

**Keywords:** QKD, distillation, amplification, reconciliation

## Abstract

We present a new post-processing method for Quantum Key Distribution (QKD) that raises cubically the secret key rate in the number of double matching detection events. In Shannon’s communication model, information is prepared at Alice’s side, and it is then intended to pass it over a noisy channel. In our approach, secret bits do not rely in Alice’s transmitted quantum bits but in Bob’s basis measurement choices. Therefore, measured bits are publicly revealed, while bases selections remain secret. Our method implements sifting, reconciliation, and amplification in a unique process, and it just requires a round iteration; no redundancy bits are sent, and there is no limit in the correctable error percentage. Moreover, this method can be implemented as a post-processing software into QKD technologies already in use.

## 1. Introduction

To put it in historical context, fiber-optic telecommunications over long distances was not possible until manufacturing techniques that improved drastically its efficiency were developed. Fibers had been used to see inside the body, but they remained unusable for long-distance information transfer because too much light was lost along the way. However, in the 1960s, Charles Kao introduced a new disruptive approach based on pure glass fibers and laser technology with transcendent achievements [1].

In the quantum era, Quantum Key Distribution (QKD) is one of the most promising technologies to secure the information intended to cross data networks. However, the development of new techniques for the rapid establishment of secret key information using quantum pulses over long distances has become unpostponable [2,3,4,5,6].

Unfortunately, some factors prevent QKD of becoming a widely used technology as its inability to reach long-distances and produce large keys at high speed. The greatest weakness of QKD technology lies in its ability to gain useful information to establish a secret key despite the noise in the quantum channel [7,8]. On the one hand, noise provides the possibility for an attacker to disguise themselves, and, on the other hand, it imposes severe difficulties to correct errors produced during transmission in order to derive two identical cryptographic keys at both sides of the quantum link [9,10]. In the case of BB84 protocol, it has been estimated that a secure key can be distilled when the quantum bit error rate (QBER) is less than 11% [11].

In the few past years, we have developed a new scheme for QKD quantum called quantum flows [12,13,14] capable of resisting challenging attacks [15,16,17,18,19,20,21,22,23,24,25]. In quantum flows approach, Alice sends to Bob a pair of quantum states, parallel or non-orthogonal, which is chosen randomly. Bob measures the two quantum states with the same measurement basis, *X* or *Z* under active basis selection. If Bob obtains the same result, a single bit has been transmitted from Alice to Bob. Quantum flows have allowed us to formulate a new method for QKD distillation based on binary structures called frames. Framed reconciliation integrates the regular QKD stages of sifting, reconciliation, and amplification in a unique process. This property makes our method unique in the context of QKD distillation; moreover, it accelerates convergence and produces a key that grows cubically in the number of double detection events.

In this work, we enhance the framed reconciliation method showed previously for 2 × 2 frames [14], and we discuss that framed reconciliation can surpass Shannon’s information bounds for noisy channels. We strongly recommend that the reader consults our previous work on Quantum Key Distillation Using Binary Frames, so that we can keep the present article concise, as far as possible. Basic concepts comprise quantum flows, non-orthogonal quantum states, quantum photonic gains, binary frames, and matching results (MR). Having introduced 2 × 2 frames, which are the frames with the minimum size, we discuss here 3 × 2 frames. Throughout the article, we will compare both schemes.

## 2. Communication Model

Classical theory of communication, as it was established by Claude Shannon in 1948, defines a general communication system where Alice (the information source) prepares an information signal, that she sends over a noisy channel, but it corrupts at least in part due to the presence of noise in the channel [26,27]. At the other side, Bob receives this information signal, but Alice and Bob must implement a processing method to recover from the errors produced during transmission [28,29,30,31,32].

Shannon’s theory imposes a limit to the highest transmission speed over a noisy channel because it can never surpass the channel capacity. The coding rate is computed as the number of message symbols divided by the number of transmitted signals. A higher coding rate means higher transmission speed. When the efficiency of the codes approximates to the channel capacity by increasing the number of transmitted signals, it is known that these codes approach to the Shannon limit. However, a coding rate too high makes it impossible to achieve a decoding error probability close to zero because the optimum channel capacity is achievable just by letting the number of transmitted signals reach infinity [33]. We claim our method goes beyond this limit because it does not require the number of transmitted signals to be increased. In fact, the coding rate reaches unity. The QKD protocol in Reference [34] exhibits a total efficiency of the communication to come up to 100%, but it does not define an error correction algorithm.

On the other side, if *e* is the probability that a transmitted 0 bit is received as a 1 and 1−e is the probability to be received as a 0, Shannon theory implies that, in case that e=0.5, one can never say anything about the original message [35,36] because the entropy is maximized when the two possible outcomes are equally probable. Since our method corrects errors when e=0.5, we claim that it goes beyond the limits implied by Shannon’s theory.

In our approach, we call active (or real) information that which is derived from Shannon’s model viewpoint because information is first prepared by Alice, then transmitted through the (quantum) channel, and, finally, recovered by Bob after it has been measured and proven to be correct. Conversely, in our scheme, information is not enclosed in the transmitted quantum pulses but in the quantum bases (*X* or *Z*) that Bob chooses at the other side. In fact, measured bits are publicly announced but the measurement bases are never revealed. We designated reactive information to this communication paradigm that we introduced to the sifting QKD procedure.

Reactive bits are computed using Bob’s measurement bases, so errors produced in the quantum channel are easily detected by Alice because such bits are publicly revealed by Bob. Remarkably, in the presence of the unit error rate, information can still be recovered since errors give reactive information by themselves. For the same reason, not all of Alice’s information can be recovered, even in the absence of errors produced by the quantum channel.

Two reconciliation approaches have been conceived in QKD: direct and reverse reconciliation. In reverse reconciliation (RR), Alice must infer Bob’s outcomes, rather than Bob guessing Alice’s encodings, known as direct reconciliation (DR). Under this classification frame, reconciliation is RR, so let us briefly contrast our approach with RR which was introduced in the context of continuous variable QKD [31,37].

It has been demonstrated that RR reconciliation achieves longer distances even beyond the 3dB limit of previous CV-QKD works [38]. RR reconciliation has been implemented over LDPC basis [39], and it was shown that LDPC codes can reach within 0.0045 dB of the Shannon limit. Unfortunately, it requires large block lengths ($10^{7}$) [40]. Even more, decoding LDPC has larger computational and memory requirements than either Cascade or Winnow algorithms [41]. In contrast, our method does not require additional bits which reduces the coding rate. Our experimental simulations show complete efficiency in detecting/correcting errors. Moreover, the secret throughput grows cubically in the number of double detection events.

Before we introduce 3 × 2 frames, we will explain quantum communication based on frames through a simple example about our reconciliation method. To facilitate its exposition, we use 2 × 2 frames in this example. Then, to simplify exposition we discuss the role of auxiliary frames in the 2 × 2 case. In Section 3, we address the research methodology for 3 × 2 frames and then we detail the QKD distillation protocol. To make the discussion more effective, we have placed tables of 3 × 2 protocol in the Appendix A. Finally, in Section 4, we analyze the efficiency and the security of the 3 × 2 protocol against different attacks as the Intercept-Resend (IR) attack and the Photon Number Splitting (PNS) attack.

### 2.1. Quantum Communication

In the BB84 protocol [42,43,44,45], a quantum state $|i_{X}\rangle$ (or $|i_{Z}\rangle$), where *i* represents the encoded bit (i=0,1), is useful to be distilled whenever it has been measured in the proper (compatible) quantum basis, basis *X* for $|i_{X}\rangle$ (or *Z* for $|i_{Z}\rangle$). Otherwise, a non-compatible measurement is produced, the bit derived from this measurement is ambiguous, and it must be discarded. However, in the quantum flows scheme, ambiguous cases can still be used for the following reasons [14]:The states are grouped by non-orthogonal pairs $\left(|i_{X}\rangle ,|i_{Z}\rangle \right)$ or $\left(|i_{X}\rangle ,|{\left(i-1\right)}_{Z}\rangle \right)$, where i=0,1.A non-orthogonal pair is measured with the same quantum basis *X* or *Z*. Both measurements yield the same result half of the times, i.e., if measuring $\left(|i_{X}\rangle ,|i_{Z}\rangle \right)$ with *X* (or *Z*) gives *i*, or measuring $\left(|i_{X}\rangle ,|{\left(i-1\right)}_{Z}\rangle \right)$ with *X* (or *Z*) gives *i* or 1−i, in both cases. We call those cases double matching detection event. Then, non-compatible measurements never occur.It implies that the bit encoded in the *X* or *Z* basis is transmitted from Alice to Bob. This communication model defines two communication channels, channel *X* and channel *Z*, because there are two bits enclosed in a non-orthogonal quantum pair: one bit over channel *X* and other bit in channel *Z*. Bob just chooses which channel he wants to use. Provided a double matching detection event is generated, both measurements are equally useful.

### 2.2. Example of Error Correction

In order to better introduce our communication model, let us illustrate it with a simple example to contrast it with Shannon’s model. To see the effect of the errors instead of the losses in the channel, let us assume a conservative quantum channel. Table 1 shows an hypothetical QKD protocol possibly based on BB84, where Alice has sent 18 quantum states (in practical implementations, some sifted bits must be sacrificed to estimate the error rate of the channel). In this example, a 30% error rate (*e*) is produced; therefore, the QKD distillation process must be declined because prominent reconciliation algorithms, such as Cascade, Winnow, or LDPC, cannot work with this high error rate.

Let us suppose that the same errors are produced using the framed reconciliation method as it is illustrated in Figure 1. In this example, we ignored the losses due to double detection events and the amplification gain produced by the amount of combinations between double matching detection events (we will discuss them later). The reconciliation based on frames can process this error rate; in fact, it can reconcile any error rate that *e* has in the channel, so there is no need to estimate *e* wasting bits for this purpose. To simplify the exposition, in this example, we used 2 × 2 frames, but we will discuss 3 × 2 frames in the Distillation Method section.

### 2.3. Auxiliary Frames

A major component of the framed reconciliation method relies in the auxiliary frames. There are two types of auxiliary frames: zero frames and testing frames. Every quantum state of a zero frame is $|0_{X}\rangle$ or $|0_{Z}\rangle$. Identifying measurement errors in a zero frame is easy, as we will see later. A testing frame contains one row that is under evaluation because it presumably contains error, and the rest of the rows come from a zero verified frame.

To compute the sifting string (SS), we follow the next procedure: A sifting string is constructed concatenating the bits that result after the ⊕ logical operation is applied to each column of the frame (a blank space is treated as a zero bit) and putting the measured bits that are produced by the optical detectors. The secret bits are derived from the code that is assigned to the arrangement of measurements inside the frame. We call measurement results (MR) to this arrangement. To see the role of auxiliary frames, let us assume that we intend to apply the framing algorithm to the Shannon’s model; thus, several zero bits are interleaved between the secret bits to be used as auxiliary correcting bits.
To achieve reconciliation in Shannon’s model, the first step is to ensure that auxiliary zero bits are error-free. However, Shannon’s 2 × 1 frames does not allow to identify errors in two consecutive zero bits (at least in one round iteration) as indicated by the following relations:$\left(\begin{array}{c} 0\\ ⊕\\ 0 \end{array}\right)=\left(\begin{array}{c} 1\\ ⊕\\ 1 \end{array}\right)=0\;\left(\mathrm{SS}\right).$In addition, when using 2 × 1 frames, there is a unique possible matching result (MR), that is written below; therefore, no secret information can be derived from MRs in Shannon’s model.$\left(\begin{array}{c} |•\rangle \\ |•\rangle \end{array}\right).$By contrast, using 2 × 2 frames, errors in the auxiliary frames can be easily identified. Here, we list the error-free zero frames:$\left(\begin{array}{c} |0_{X}\rangle \;-\\ ⊕\\ -\;|0_{Z}\rangle \end{array}\right)=\left(\begin{array}{c} -\;|0_{Z}\rangle \\ ⊕\\ |0_{X}\rangle \;- \end{array}\right)=\left(\begin{array}{c} |0_{X}\rangle \;-\\ ⊕\\ |0_{X}\rangle \;- \end{array}\right)=\left(\begin{array}{c} -\;|0_{Z}\rangle \\ ⊕\\ -\;|0_{Z}\rangle \end{array}\right)=00,00\;\left(\mathrm{SS}\right),$which can be compared, for illustrative purposes, to the erroneous cases:$\left(\begin{array}{c} |1_{X}\rangle \;-\\ ⊕\\ -\;|1_{Z}\rangle \end{array}\right)=\left(\begin{array}{c} -\;|1_{Z}\rangle \\ ⊕\\ |1_{X}\rangle \;- \end{array}\right)=11,11\;\left(\mathrm{SS}\right),$$\left(\begin{array}{c} |1_{X}\rangle \;-\\ ⊕\\ |1_{X}\rangle \;- \end{array}\right)=\left(\begin{array}{c} -\;|1_{Z}\rangle \\ ⊕\\ -\;|1_{Z}\rangle \end{array}\right)=00,11\;\left(\mathrm{SS}\right).$Ambiguous SS are produced in regular frames. For example, to the left, we indicate that Alice sends the frame $f_{2}$ to Bob, who measures it using MR = 11. However, when applying the *Z* measurement basis, the photo-detector yields an error reporting $|1_{Z}\rangle$ instead $|0_{Z}\rangle$; so, we have:$f_{2_{a}}=$$\left(\begin{array}{cc} |1_{X}\rangle \; & |0_{Z}\rangle \\ |1_{X}\rangle \; & |1_{Z}\rangle \end{array}\right),$$f_{2_{b}}=$$\left(\begin{array}{c} -\;|1_{Z}\rangle \\ ⊕\\ |1_{X}\rangle \;- \end{array}\right)=11,11\;\left(\mathrm{SS}\right).$When Alice receives the string SS = 11,11 which belongs to $f_{2}$, she knows it implies two possibilities: either SS comes from the error-free string ${\mathrm{SS}}_{24}=11,11$ under MR = 10 in $f_{2}$ or an error is produced in the first measured bit that actually corresponds to the string ${\mathrm{SS}}_{23}=10,01$ under MR = 11 in $f_{2}$. To disambiguate it, Alice uses the auxiliary frame $f_{10}$. Thus, she looks at a frame $f_{10}$ where the ambiguous row $\left(-,|1_{Z}\rangle \right)$ is allocated. Remember that each row is combined with each other. Previously, the second row of $f_{10}$, i.e., $\left(|0_{X}\rangle ,-\right)$, was verified as a zero frame. Then, suppose Alice finds the following $f_{10}$ case:$f_{10}=$$\left(\begin{array}{c} -\;|1_{Z}\rangle \\ ⊕\\ |0_{X}\rangle \;- \end{array}\right)=10,10.$

The sifting string 10,10 reveals that an error exists in the row that is under evaluation; therefore, Alice decides ${\mathrm{SS}}_{23}$. Then, the pair (${\mathrm{SS}}_{23}$,$f_{2}$) determines Alice’s secret bit. It must be highlighted that the sifting strings of auxiliary frames cannot be distinguished from other identical SS from regular frames, so privacy is guaranteed. In fact, it is ensured that each SS can proceed equally from each bit.

### 2.4. One-Time Pad XOR Equivalency

It is known that the XOR one-time pad encryption method is a perfect cryptosystem provided the crypto key achieves the same number of bits as the plaintext. Let us show that the framing method actually behaves as one-time encryption. First, in Table 4, we can see the logical XOR (⊕) function. Each encrypted bit *c* could be produced by each key bit denoted as *k*.

As specified in the framed reconciliation method [14], Bob must reveal the sifting bits along the measured bits. However, each SS maps two different MRs, as can be verified in Table 5. Since secret bits are enclosed in MRs, we proved that secret bits of the framing protocol are equivalent to the secret bits of the XOR one-time pad cryptosystem. The same analysis can be applied to the 3 × 2 frames.

## 3. Distillation Method with 3 × 2 Frames

Before we detail the steps of the distillation method for 3 × 2 frames, let us describe the research methodology we applied:The 3 × 2 frames must be identified: there are $4^{3}=64$ binary 3 × 2 frames.The measurement results (MR) must be specified: in 3 × 2 frames, there are 8 MR. Those MR are illustrated in Table A2 of Appendix A.Frames are classified as usable and useless frames: a usable frame is a frame that produces a distinct SS under each MR. In 3 × 2 frames, there are 8 distinct SS per frame and 24 usable frames. Sifting bits are written in Table A4 of Appendix A. Remember that Sifting Strings (SS) are composed by the sifting bits and the measured bits: SS=1stsiftingbit||2ndsiftingbit||3thsiftingbit,1stmeasuredbit||2ndmeasuredbit||3thmeasuredbit. The 3thsiftingbit is appended to achieve discrimination, and it can be considered as a parity sifting bit.Auxiliary frames which are intended to catch errors produced in regular frames must be identified. In 3 × 2 frames, there are 3 auxiliary frames labeled as $f_{25}$, $f_{26}$, and $f_{27}$. The frame $f_{25}$ is the zero frame and is used to verify the two (below) rows of the testing frames $f_{26}$ and $f_{27}$. The upper row of $f_{26}$ and $f_{27}$ is the row that is being tested. In the end, Alice will include the auxiliary frames inside the set of frames that Bob must remove. Auxiliary frames are listed in Table A1 of Appendix A.All usable frames under each MR must be expanded to analyze all possible errors through SS, from single to multiple errors. Then, ambiguous SS that can be corrected under the auxiliary frames must be detected. In addition, all the SS that cannot be disambiguated must be identified and the corresponding frames must be removed. We show in Table A5 the cases that can be successfully disambiguated.At Bob’s side, each (SS, MR) pair defines a secret bit (sb). For Alice, the same secret bit results from the pair (SS, $f_{i}$) because she knows the frame that is behind each SS. It must be guaranteed that each SS can be produced equally by both bits. In addition, it must be ensured that each secret bit proceeds from the same number of frames, so that the bit probability of each SS is the same in order to reduce the eavesdropper’s information gain (SS are publicly transmitted over the classical channel). This action may involve removing some extra SS. Alice sends to Bob the set of SS of all the frames that must be eliminated including auxiliary frames. Table A3 of Appendix A enlists SS, MR, frames, and sb.

Now, we can proceed to summarize the steps of the distillation method for 3 × 2 frames that comprises sifting, reconciliation, and privacy amplification. The overall steps of the process are indicated in Figure 2:Alice sends some non-orthogonal quantum pairs either $\left(|i_{X}\rangle ,|i_{Z}\rangle \right)$ or $\left(|i_{X}\rangle ,|{\left(1-i\right)}_{Z}\rangle \right)$ where i=0,1. Although quantum non-orthogonal pairs can be mutually interleaved they are numbered, so each pair can be identified by Alice and BobBob measures each quantum pair using the same measurement basis (*X* or *Z*) which is chosen randomly (under active basis measurement). Some double detection events are produced. Bob informs Alice the tag number of such quantum pairs.Alice computes all usable frames including null frames and auxiliary frames. She communicates to Bob the frame arrangement information. We call this step privacy amplification.Bob computes the Sifting String (SS) of each frame. He returns the set of Siting Strings he obtained to Alice.Alice analyzes the SS received from Bob: She generates frames $f_{25}$ to prepare the auxiliary frames.Using auxiliary frames, Alice removes ambiguity. Alice gets the secret bits using the relation (SS, $f_{i}$) and Table A3 of Appendix A. Alice informs Bob of the cases that must be eliminated (because they cannot be disambiguated).Bob removes the frames identified by Alice to reach Alice’s secret bit string. Bob’s secret bits are derived from (SS, MR) and Table A3 of Appendix A.

## 4. Secret Rate

The secret rate of the framed reconciliation method can be derived directly from frames without recurring to quantum physics mathematical relations. First off, we must enlist the Sifting String (SS) generated by all the frames classified by Measurement Result (MR) and separate the error-free SS from the erroneous SS (single and multiple errors). According to the size of frames (2 × 2 or 3 × 2), the error could be in the first bit, second bit, third bit, two bits, two of three bits, and three bits simultaneously. Then, we proceed to identify ambiguous SS, (because they appear simultaneously as error-free SS and erroneous SS for a given frame). Then, we identify the SS that can still be used after they are inspected under auxiliary frames. We call those cases unequivocal SS cases.

We calculate the secret rate (in absence of eavesdropping) as the sum up of the information derived from the unequivocal error-free rate and the amount of information derived from the unequivocal erroneous rate (unequivocal error-free rate is obtained as the number of unequivocal error-free SS under the total number of error-free SS; conversely, the unequivocal error rate is obtained as the rate of unequivocal erroneous SS over the total erroneous SS cases). As mentioned earlier, unequivocal means that ambiguity can be removed using auxiliary frames. The bits from remaining SS must be eliminated since they do not contribute to the secret rate.

In Table 6, we detail the deduction of the secret rate. Each SS contributes with a single bit. In 2 × 2 frames, we have 4 usable frames, and each one generates 4 SS; to compute the unequivocal erroneous rate, we have 2 SS per frame that can be recovered from 12 SS per frame yields $\frac{1}{6}$. On the other hand, to derive the unequivocal error-free rate, we have 2 SS per frame that can be recovered from 4 SS per frame it yields $\frac{1}{2}$. The unequivocal erroneous rate in 3 × 2 frames yields $\frac{1}{3}$, and the unequivocal error-free rate gives $\frac{1}{21}$ (see Figure 3).

### 4.1. Secret Throughput

One of the main advantages of the reconciliation method based on frames is the total number of secret bits that results when the framing gain is applied. Remarkably, framing gain results from the amount of total combinations among double matching detection events. We call this process privacy pre-amplification (or amplification in short). Therefore, we compute the secret throughput multiplying the secret rate by the framing gain. In the case of 2 × 2 frames, we have 4 usable frames under 16 total frames, so the framing gain is 14n2. Conversely, in 3 × 2 frames, there are 24 over 64 frames, so the framing gain is 38n3. Equation (2) describes the secret throughput for each case.
(1)Iab(2x2)=14n212−13eIab(3x2)=38n313−27e.

Just to appreciate the growth rate of each frame size, we compute, in Table 7, some values of the secret throughput as a function of *n* and *e*. As it can be inferred, 3 × 2 frames have a visible advantage to produce secret bits, e.g., when $n=10^{3}$, it raises the secret throughput to $n=10^{8}$ bits.

### 4.2. Rate Code

The rate code $r_{ab}$ is the relation between the secret information and the total bits generated to achieve reconciliation. In the case of 2 × 2 frames, the total information is 4n2, while the total number is 6n3 in 3 × 2 frames. The rate code for each size of frame is written in Equation (2).
(2)$$\begin{array}{cc} r_{ab_{\left(2\times 2\right)}} & =\frac{1}{16}\left(\frac{1}{2}-\frac{1}{3}e\right)\\ r_{ab_{\left(3\times 2\right)}} & =\frac{1}{16}\left(\frac{1}{3}-\frac{2}{7}e\right) \end{array}.$$

### 4.3. Secret Key Rate

In the case of frame reconciliation, the eavesdropper has a great disadvantage since they do not know Bob’s bases selection because they are not revealed over the classical channel. Even if the eavesdropper captures some copies of the quantum pulses, they must deal with the double detection events and the basis choices. Moreover, although the eavesdropper could replicate some double detection events, Alice performs all combinations between double detection events. As a consequence of the privacy amplification process, the eavesdropper’s information reduces even more.

#### 4.3.1. The Intercept and Resend Attack (IR)

In the Intercept and Resend (IR) attack, the eavesdropper first measures each pair of non-orthogonal quantum pulses in the quantum channel, and then they send another pair of quantum pulses to Bob prepared according to the same quantum states.

Since secret bits are derived only from double matching detection events, Eve must produce first a double matching detection event using the quantum states she intercepts in the quantum channel because no useful information could be extracted from double non-matching detection events nor even single detection events.

In addition, Eve must guarantee that both states she resends to Bob’s station achieve his optical detectors, which imposes a severe difficulty because vacuum or single detection events are more probable than double detection events. However, suppose Eve forces both quantum states to arrive Bob’s receiver station. We can derive the efficiency of the IR attack using the following example:—Alice sends the non-orthogonal pair $\left(|0_{X}\rangle ,|0_{Z}\rangle \right)$ to Bob over the quantum channel. Eve measures them using *Z* basis, and let us assume she obtains a double matching detection event, say $\left(|0_{Z}\rangle ,|0_{Z}\rangle \right)$.—Eve prepares and sends the quantum pair $\left(|0_{Z}\rangle ,|0_{Z}\rangle \right)$ to Bob.—Suppose Eve can force both quantum pulses to arrive to Bob’s optical station. There are two quantum measurement bases (*X* or *Z*) and five possible outcomes:
–$\frac{1}{2}$ due to Bob’s *Z* basis: $\left(|0_{Z}\rangle ,|0_{Z}\rangle \right)$.–$\frac{1}{2}$ due to Bob’s *X* basis: $\left\{(|0_{X}\rangle ,|0_{X}\rangle ),(|1_{X}\rangle ,|1_{X}\rangle ),(|1_{X}\rangle ,|0_{X}\rangle ),(|0_{X}\rangle ,|1_{X}\rangle )\right\}$.To match Eve’s double detection event $\left(|0_{Z}\rangle ,|0_{Z}\rangle \right)$, Bob must choose the *Z* basis which occurs with $\frac{1}{2}$ probability, so Eve’s final probability is $\frac{1}{4}$.

The overall scheme is depicted in the following diagram, where $Q_{\left(+,+\right)}$ represents Alice’s pairs of non-orthogonal states:



#### 4.3.2. The Photon Number Splitting Attack (PNS)

The eavesdropper has a copy of all the quantum states that arrive to Bob’s station because Alice sends attenuated (multi-photon) quantum pulses, and the eavesdropper is equipped with a sufficiently large quantum memory. However, the eavesdropper’s probability of getting a double matching detection event is $\frac{1}{2}$. In addition, Eve must measure choosing between two different measurement basis (*X* or *Z*); thus, his final probability is $\frac{1}{4}$:$\frac{1}{2}$ because of the probability to get a double matching detection event.$\frac{1}{2}$ due to basis matching. Eve must measure choosing between two different measurement basis (*X* or *Z*).



#### 4.3.3. The Bases Choice Attack (BC)

The eavesdropper would decide to apply another quantum measurement bases to gain more information, and then they use the measurement bases X+Z or X−Z. First, consider that the eavesdropper chooses between the measurement bases (X+Z or X−Z) with 0.5 probability. However, non-matching detection events are ambiguous for the eavesdropper, which occur with $\frac{6}{16}$ probability. In contrast, they get a double matching event with $\frac{9}{16}$ probability. As a result, the chance to get Bob’s information is $\frac{9}{32}$.

Equation (Equation 3) shows the relation to compute the secret key rate for each frame size. It is written as the secret information multiplied by the rate between the total frames produced by Alice and those the eavesdropper duplicates.
(3)ΔI(2X2)=12−13e1−R·n2n2ΔI(3X2)=13−27e1−R·n3n3.

Table 8 shows the final secret key information for each attack: Intercept and Resend attack (IR), Photon Number Splitting attack (PNS), and Basis Choice attack (BC). In the case of 2 × 2 frames, we have ignored the linear term *n* that is generated in n2 because the quadratic term $n^{2}$ is dominant. In the same way, we omitted the quadratic and linear terms produced by n3 because of the high order of the cubic term.

As it can be deduced from Table 8, the secret key rate is affected slightly by the eavesdropper’s behavior. This new scenario opens the possibility to employ less attenuated pulses as in CV-QKD to achieve, on one hand, long-distances quantum links or, on the other, portable QKD in closed buildings [46].

## 5. Conclusions

We have discussed a new post-processing method for Quantum Key Distribution (QKD) that raises cubically the secret key rate in the number of double matching detection events. Secret bits are derived from reactive bits instead of Shannon information, so Bob’s measured bits are publicly revealed, while bases selections remain secret. Our method implements sifting, reconciliation, and amplification in a unique process, and it just requires a round iteration; no redundancy bits are sent, and no limit in the correctable error percentage. Despite the fact that the reconciliation is performed with a unity error channel, the secret rate is kept, at least theoretically, in 16% using 2 × 2 frames and 4.7% when using 3 × 2 frames.

It is not difficult to evaluate the security of this method because it can be evaluated directly through the frames. There is no dependency on other security mechanism as hash functions.

The protocol works fast, at least theoretically, convergence is guaranteed, and it can be implemented as a post-processing software into QKD technologies.

## Figures and Tables

**Figure 1 entropy-23-00229-f001:**
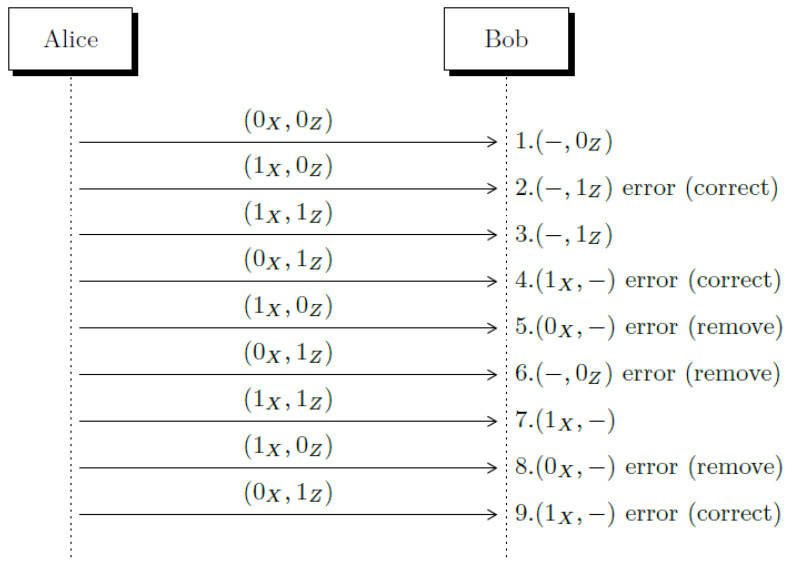
Using frame reconciliation, all errors are detected and corrected (or removed). Each double detection event has been enumerated to follow them into the frames (see Table 2 and Table 3).

**Figure 2 entropy-23-00229-f002:**
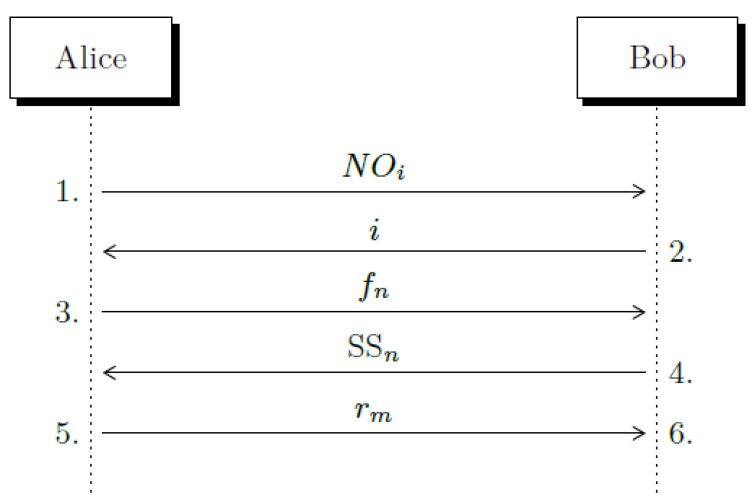
The frame distillation runs in one iteration: Alice sends pairs of non-orthogonal states ($NO_{i}$). Bob informs to Alice which cases produced double matching detection events (*i*). Alice generates all possible frames and sends to Bob the frame arrangement information ($f_{n}$). Bob returns back the sifting strings (SS_*n*_). Finally, Alice tells Bob which cases he must delete ($r_{m}$). Step 1 is executed over the quantum channel, while steps 2 to 5 are completed using the classical channel.

**Figure 3 entropy-23-00229-f003:**
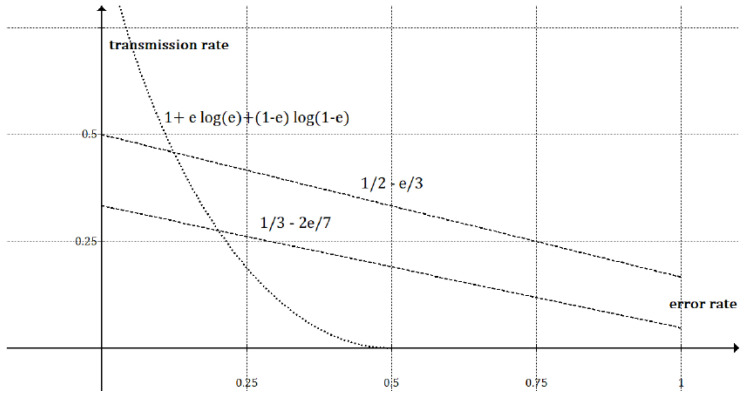
The theoretical transmission rate is plotted as a function of the quantum bit error rate (QBER) *e*; we show the 2 × 2 and 3 × 2 lines and the Shannon’s reference function. When e=1, the secret rate achieves 0.16 for 2 × 2 frames and 0.047 for 3 × 2 frames.

**Table 1 entropy-23-00229-t001:** In this example of a running Quantum Key Distribution (QKD), 6 errors (underlined at Bob’s column) among 18 measured quantum states are produced, so it gives an error rate of 30%. According to Shannon’s limit, it yields a transmission rate of 0.0817. It is known that, at 50%, there is no reconcilable information.

Alice	Bob
${|0_{X}\rangle }_{2}$, ${|0_{Z}\rangle }_{1}$,	${|0_{X}\rangle }_{2}$, ${|0_{Z}\rangle }_{1}$,
${|1_{X}\rangle }_{4}$, ${|0_{Z}\rangle }_{3}$,	${|1_{X}\rangle }_{4}$, $\underline{{|1_{Z}\rangle }_{3}}$,
${|1_{X}\rangle }_{6}$, ${|1_{Z}\rangle }_{5}$,	${|1_{X}\rangle }_{6}$, ${|1_{Z}\rangle }_{5}$,
${|0_{X}\rangle }_{8}$, ${|1_{Z}\rangle }_{7}$,	$\underline{{|1_{X}\rangle }_{8}}$, ${|1_{Z}\rangle }_{7}$,
${|1_{X}\rangle }_{10}$, ${|0_{Z}\rangle }_{9}$,	$\underline{{|0_{X}\rangle }_{10}}$, ${|0_{Z}\rangle }_{9}$,
${|0_{X}\rangle }_{12}$, ${|1_{Z}\rangle }_{11}$,	${|0_{X}\rangle }_{12}$, $\underline{{|0_{Z}\rangle }_{11}}$,
${|1_{X}\rangle }_{14}$, ${|1_{Z}\rangle }_{13}$,	${|1_{X}\rangle }_{14}$, ${|1_{Z}\rangle }_{13}$,
${|1_{X}\rangle }_{16}$, ${|0_{Z}\rangle }_{15}$,	$\underline{{|0_{X}\rangle }_{16}}$, ${|0_{Z}\rangle }_{15}$,
${|0_{X}\rangle }_{18}$, ${|1_{Z}\rangle }_{17}$	$\underline{{|1_{X}\rangle }_{18}}$, ${|1_{Z}\rangle }_{17}$

**Table 2 entropy-23-00229-t002:** Alice receives the Sifting String (SS) from Bob, which she knows belongs to $f_{2}$, $f_{3}$, and $f_{4}$, respectively, but they are ambiguous, so she uses the auxiliary frames $f_{10}$, $f_{9}$, and $f_{9}$, respectively, to identify the error and then correct it.

$\begin{array}{c} f_{2} \end{array}\begin{array}{c} 2.\;\\ 3.\; \end{array}\begin{array}{c} \mathrm{MR}\;=\;01\\ \left(\begin{array}{cc} \;-\; & |1_{Z}\rangle \\ \;-\; & \;|1_{Z}\rangle \; \end{array}\right)\\ \mathrm{SS}\;=\;00,11 \end{array}$	$\begin{array}{c} f_{10} \end{array}$ $\begin{array}{c} 2.\;\\ 1.\; \end{array}$ $\begin{array}{c} \mathrm{MR}\;=\;01\\ \left(\begin{array}{cc} \;-\; & |1_{Z}\rangle \\ \;-\; & \;|0_{Z}\rangle \; \end{array}\right)\\ \mathrm{SS}\;=\;01,10 \end{array}$
$\begin{array}{c} f_{3} \end{array}$ $\begin{array}{c} 4.\;\\ 3.\; \end{array}$ $\begin{array}{c} \mathrm{MR}\;=\;10\\ \left(\begin{array}{cc} |1_{X}\rangle & -\\ - & |1_{Z}\rangle \end{array}\right)\\ \mathrm{SS}\;=\;11,11 \end{array}$	$\begin{array}{c} f_{9} \end{array}$ $\begin{array}{c} 4.\;\\ 1.\; \end{array}$ $\begin{array}{c} \mathrm{MR}\;=\;10\\ \left(\begin{array}{cc} |1_{X}\rangle & -\\ - & |0_{Z}\rangle \end{array}\right)\\ \mathrm{SS}\;=\;10,10 \end{array}$
$\begin{array}{c} f_{4} \end{array}$ $\begin{array}{c} 7.\;\\ 9.\; \end{array}$ $\begin{array}{c} \mathrm{MR}\;=\;00\\ \left(\begin{array}{cc} |1_{X}\rangle & \;-\;\\ |1_{X}\rangle & \;-\; \end{array}\right)\\ \mathrm{SS}\;=\;00,11 \end{array}$	$\begin{array}{c} f_{9} \end{array}$ $\begin{array}{c} 9.\;\\ 1.\; \end{array}$ $\begin{array}{c} \mathrm{MR}\;=\;10\\ \left(\begin{array}{cc} |1_{X}\rangle & -\\ - & |0_{Z}\rangle \end{array}\right)\\ \mathrm{SS}\;=\;10,10 \end{array}$

**Table 3 entropy-23-00229-t003:** After Alice receives these SS, she determines that the respective frames must be eliminated because ambiguity cannot be removed.

$\begin{array}{c} f_{2} \end{array}$ $\begin{array}{c} 5.\;\\ 3.\; \end{array}$ $\begin{array}{c} \mathrm{MR}\;=\;01\\ \left(\begin{array}{cc} |0_{X}\rangle & -\\ - & |1_{Z}\rangle \end{array}\right)\\ \mathrm{SS}\;=\;01,01 \end{array}$	$\begin{array}{c} f_{3} \end{array}$ $\begin{array}{c} 6.\;\\ 3.\; \end{array}$ $\begin{array}{c} \mathrm{MR}\;=\;01\\ \left(\begin{array}{cc} -\; & \;|0_{Z}\rangle \\ \;-\; & \;|1_{Z}\rangle \end{array}\right)\\ \mathrm{SS}\;=\;01,01 \end{array}$	$\begin{array}{c} f_{6} \end{array}$ $\begin{array}{c} 8.\;\\ 7.\; \end{array}$ $\begin{array}{c} \mathrm{MR}\;=\;01\\ \left(\begin{array}{cc} |0_{X}\rangle & \;-\;\\ \;|1_{X}\rangle \; & \;-\; \end{array}\right)\\ \mathrm{SS}\;=\;10,01 \end{array}$

**Table 4 entropy-23-00229-t004:** The logical XOR function.

*c*	k⊕b
0	0⊕0
	1⊕1
1	0⊕1
	1⊕0

**Table 5 entropy-23-00229-t005:** The XOR function for 2 × 2 frames; matching results (MR) is the measurement result, and sb denotes the final secret bit.

c	k⊕b	MR	Frames	sb
00	$\left(|0_{X}\rangle ,-\right)⊕\left(-,|0_{Z}\rangle \right)$	10	$f_{1}$	0
	$\left(-,|0_{Z}\rangle \right)⊕\left(|0_{X}\rangle ,-\right)$	11	$f_{5}$	1
	$\left(|1_{X}\rangle ,-\right)⊕\left(|1_{X}\rangle ,-\right)$	00	$f_{2}$, $f_{6}$	0
	$\left(-,|1_{Z}\rangle \right)⊕\left(-,|1_{Z}\rangle \right)$	01	$f_{3}$, $f_{4}$	1
01	$\left(-,|1_{Z}\rangle \right)⊕\left(-,|0_{Z}\rangle \right)$	01	$f_{1}$, $f_{6}$	0
	$\left(-,|1_{Z}\rangle \right)⊕\left(|0_{X}\rangle ,-\right)$	11	$f_{4}$	1
	$\left(|0_{X}\rangle ,-\right)⊕\left(-,|1_{Z}\rangle \right)$	10	$f_{3}$	0
	$\left(-,|0_{Z}\rangle \right)⊕\left(-,|1_{Z}\rangle \right)$	01	$f_{2}$, $f_{5}$	1
10	$\left(|1_{X}\rangle ,-\right)⊕\left(|0_{X}\rangle ,-\right)$	00	$f_{4}$, $f_{5}$	0
	$\left(|1_{X}\rangle ,-\right)⊕\left(-,|0_{Z}\rangle \right)$	10	$f_{6}$	1
	$\left(|0_{X}\rangle ,-\right)⊕\left(|1_{X}\rangle ,-\right)$	00	$f_{1}$, $f_{3}$	0
	$\left(-,|0_{Z}\rangle \right)⊕\left(|1_{X}\rangle ,-\right)$	11	$f_{2}$	1
11	$\left(-,|1_{Z}\rangle \right)⊕\left(|1_{X}\rangle ,-\right)$	11	$f_{1}$, $f_{3}$, $f_{6}$	0
	$\left(|1_{X}\rangle ,-\right)⊕\left(-,|1_{Z}\rangle \right)$	10	$f_{2}$, $f_{4}$, $f_{5}$	1

**Table 6 entropy-23-00229-t006:** The secret rate is indicated without taking the framing gain for each frame size. The secret rate is shown when e=0 and e=1.

$I_{ab_{\left(2\times 2\right)}}$	$I_{ab_{\left(3\times 2\right)}}$
$\frac{1}{2}\left(1-e\right)+\frac{1}{6}e$	$\frac{1}{3}\left(1-e\right)+\frac{1}{21}e$
$\frac{1}{2}-\frac{1}{3}e$	$\frac{1}{3}-\frac{2}{7}e$
$e=0\to I_{ab_{\left(2\times 2\right)}}=\frac{1}{2}$	$e=0\to I_{ab_{\left(3\times 2\right)}}=\frac{1}{3}$
$e=1\to I_{ab_{\left(2\times 2\right)}}=\frac{1}{6}$	$e=1\to I_{ab_{\left(3\times 2\right)}}=\frac{1}{21}$

**Table 7 entropy-23-00229-t007:** The theoretical secret throughput (bits) as a function of *n* and *e* for each frame size.

*n*	e=0		e=0.5		e=1	
	$I_{{\mathrm{ab}}_{\left(2\times 2\right)}}$	$I_{{\mathrm{ab}}_{\left(3\times 2\right)}}$	$I_{{\mathrm{ab}}_{\left(2\times 2\right)}}$	$I_{{\mathrm{ab}}_{\left(3\times 2\right)}}$	$I_{{\mathrm{ab}}_{\left(2\times 2\right)}}$	$I_{{\mathrm{ab}}_{\left(3\times 2\right)}}$
100	618	20,212	412	11,550	206	2887
500	15,593	2,588,562	10,395	1,479,178	5197	369,794
1000	62,437	20,770,875	41,625	11,869,071	20,812	2,967,267

**Table 8 entropy-23-00229-t008:** The secret key rate is computed as $\Delta I=I_{ab}-I_{ae}$ for each attack.

IR	PNS	BC
$\left(1-{\left(\frac{1}{4}\right)}^{2}\right)\;\cdot \;I_{ab_{\left(2\times 2\right)}}$	$\left(1-{\left(\frac{1}{4}\right)}^{2}\right)\;\cdot \;I_{ab_{\left(2\times 2\right)}}$	$\left(1-{\left(\frac{9}{32}\right)}^{2}\right)\;\cdot \;I_{ab_{\left(2\times 2\right)}}$
$\left(1-{\left(\frac{1}{4}\right)}^{3}\right)\;\cdot \;I_{ab_{\left(3\times 2\right)}}$	$\left(1-{\left(\frac{1}{4}\right)}^{3}\right)\;\cdot \;I_{ab_{\left(3\times 2\right)}}$	$\left(1-{\left(\frac{9}{32}\right)}^{3}\right)\;\cdot \;I_{ab_{\left(3\times 2\right)}}$

## Data Availability

The data presented in this study are available within the article.

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
