# Peer review of "Beyond the Limits of Shannon’s Information in Quantum Key Distribution"

_entropy, 2021, doi:10.3390/e23020229_

Round 1
Reviewer 1 Report
The authors present a new post-processing method for QKD protocols relying on the bases choices on the receiver side of the protocol. The method promises to be useful for QKD and can theoretically increase the secret key rates. I therefore recommend the paper for publication after minor revisions. I particular, the references on the recent review articles in the field [Rev. Mod. Phys. 92, 025002 (2020); Adv. Opt. Photon. 12, 1012-1236 (2020)] can be added to guide a reader. Then, to put the work in the context, the authors might comment on how their methods are different from the reverse reconciliation approach in continuous-variable QKD [Nature 421, 238 (2003)], where the remote trusted party is a also a reference side for the error correction. The statement "Shannon theory implies that above 50% in the error rate of the channel is impossible that Bob recovers Alice’s information" has to be supported by a relevant citation or a derivation. Finally, the authors may comment on what can be the practical challenges of implementing the suggested method.
Author Response
The authors deeply value the comprehensive comments and suggestions to perform this revision. We are very thankful to the Reviewer 1 for recognizing the importance of this research and recommending a revision to improve the quality. We have carefully followed his advice and done our best effort to address his comments. Please find attached our reply to the points raised and the corresponding modifications we have made in the revised manuscript.

Reviewer 2 Report
This manuscript proposed a novel quantum key distribution (QKD) protocol using the non-orthogonal qubit pairs. The measurement results of the qubits are disclosed by the receiver but the basis selection remains secret to the public. The authors claimed that the key rate of their protocol has surpassed Shannon’s communication model based on the number of double matching detection events. However, I found some critical mistakes in both the protocol description and the security analysis that contradict the authors' claim. Since only the basis selection remains secret in the protocol, the qubit efficiency for sharing one secret bit per qubit pair is equivalent to some classic QKD protocols [1-3] with the decoy photons and public discussions. On the other hand, each non-orthogonal qubit pair with the first quibt as X-basis, and the second qubit as Z-basis has only four combinations which leads to Eve has a quarter chance (not one-fifth as claimed by the authors) to resend the correct qubit pair during the protocol. Therefore, for the above reasons, I do not recommend this manuscript for publication in Entropy.
[1] Li, Jian, Na Li, Lei-Lei Li, and Tao Wang. "One step quantum key distribution based on EPR entanglement." Scientific reports 6 (2016): 28767.
[2] Yuan, Hao, Jun Song, Lian-fang Han, Kui Hou, and Shou-hua Shi. "Improving the total efficiency of quantum key distribution by comparing Bell states." Optics communications 281, no. 18 (2008): 4803-4806.
[3] Zhang, Zhan-Jun, Zhong-Xiao Man, and Shou-Hua Shi. "An efficient multiparty quantum key distribution scheme." International Journal of Quantum Information 3, no. 03 (2005): 555-560.
Author Response
The authors highly value the comprehensive comments and suggestions from this reviewer, as well as the time spent reviewing the manuscript. Regarding the comment: "On the other hand, each non-orthogonal qubit pair with the first quibt as X-basis, and the second qubit as Z-basis has only four combinations which leads to Eve has a quarter chance (not one-fifth as claimed by the authors) to resend the correct qubit pair during the protocol." the reviewer is describing the PNS attack which is discussed in the manuscript.
Reviewer 3 Report
This is a new article from a group that already for some years has been developing protocols to tackle practical drawbacks in the implementation of QKD protocols. They propose a method which they call Q, for quantum flows, to resist attacks in quantum communications. The idea start with the use of interleaved photons and evolved to the use of binary frames which consist in 2x2 matrices to perform the quantum distillation. In the present work they explore the possibility of 3x2 frames and the results obtained are promising. The article is well written and I recommend it for publication.
Author Response
The authors deeply value the comprehensive comments and suggestions to perform this revision. We are very thankful to the Reviewer 3 for recognizing the importance of this research and recommending the manuscript for publication.
Round 2
Reviewer 2 Report
Unfortunately, I did not see from the revised manuscript that the authors address both of my questions I raised in the previous review. They avoid the issue of the qubit efficiency of their proposed QKD protocol is equivalent to other existing QKD protocols [1-3]. Also, they redirect my question on the IR attack to the PNS attack which I am pretty sure I was talking about the IR attack. Again, since we have already known in their protocol that the first qubit is in X-basis and the second qubit is in Z-basis, Eve has a quarter chance to resend the correct qubit pair. That is, she only needs to guess the information is 0 (or 1) for each qubit. Therefore, I still do not recommend this manuscript for publication unless the authors tried to clarify my questions.
[1] Li, Jian, Na Li, Lei-Lei Li, and Tao Wang. "One step quantum key distribution based on EPR entanglement." Scientific reports 6 (2016): 28767.
[2] Yuan, Hao, Jun Song, Lian-fang Han, Kui Hou, and Shou-hua Shi. "Improving the total efficiency of quantum key distribution by comparing Bell states." Optics communications 281, no. 18 (2008): 4803-4806.
[3] Zhang, Zhan-Jun, Zhong-Xiao Man, and Shou-Hua Shi. "An efficient multiparty quantum key distribution scheme." International Journal of Quantum Information 3, no. 03 (2005): 555-560.
Author Response
We sincerely hope that the comments and suggestions of Reviewer #2 have been sufficiently addressed. We apologize for the late response to his suggestions but these have now been considered and we are sure that they have helped a lot to improve the manuscript. In addition, a thorough review of the English style of the manuscript has been performed. For the above reasons, we hope the article will be published in the Special Issue of Entropy.

Round 3
Reviewer 2 Report
I can see the efforts taken by the authors in revising this manuscript. In this revision, I can also see the authors sincerely addressed my questions. After reconsideration, I would like to recommend this paper for publication in Entropy.